# ENERGY-BASED MODELS FOR ATOMIC-RESOLUTION PROTEIN CONFORMATIONS

**Yilun Du**[*]
Massachusetts Institute of Technology
Cambridge, MA
yilundu@mit.edu

**Joshua Meier**
Facebook AI Research
New York, NY
jmeier@fb.com

**Jerry Ma**
Facebook AI Research
Menlo Park, CA
maj@fb.com

**Rob Fergus**
Facebook AI Research & New York University
New York, NY
robfergus@fb.com

**Alexander Rives**
New York University
New York, NY
arives@cs.nyu.edu

## ABSTRACT

We propose an energy-based model (EBM) of protein conformations that operates at atomic scale. The model is trained solely on crystallized protein data. By contrast, existing approaches for scoring conformations use energy functions that incorporate knowledge of physical principles and features that are the complex product of several decades of research and tuning. To evaluate the model, we benchmark on the rotamer recovery task, the problem of predicting the conformation of a side chain from its context within a protein structure, which has been used to evaluate energy functions for protein design. The model achieves performance close to that of the Rosetta energy function, a state-of-the-art method widely used in protein structure prediction and design. An investigation of the model's outputs and hidden representations finds that it captures physicochemical properties relevant to protein energy.

## 1 INTRODUCTION

Methods for the rational design of proteins make use of complex energy functions that approximate the physical forces that determine protein conformations (Cornell et al., 1995; Jorgensen et al., 1996; MacKerell Jr et al., 1998), incorporating knowledge about statistical patterns in databases of protein crystal structures (Boas & Harbury, 2007). The physical approximations and knowledge-derived features that are included in protein design energy functions have been developed over decades, building on results from a large community of researchers (Alford et al., 2017).

In this work[1], we investigate learning an energy function for protein conformations directly from protein crystal structure data. To this end, we propose an energy-based model using the Transformer architecture (Vaswani et al., 2017), that accepts as inputs sets of atoms and computes an energy for their configuration. Our work is a logical extension of statistical potential methods (Tanaka & Scheraga, 1976; Sippl, 1990; Lazaridis & Karplus, 2000) that fit energetic terms from data, which, in combination with physically motivated force

---

[*]Work performed during an internship at Facebook

[1]Data and code for experiments are available at https://github.com/facebookresearch/protein-ebm

fields, have contributed to the feasibility of *de novo* design of protein structures and functions (Kuhlman et al., 2003; Ambroggio & Kuhlman, 2006; Jiang et al., 2008; King et al., 2014).

To date, energy functions for protein design have incorporated extensive feature engineering, encoding knowledge of physical and biochemical principles (Boas & Harbury, 2007; Alford et al., 2017). Learning from data can circumvent the process of developing knowledge-based potential functions by automatically discovering features that contribute to the protein's energy, including terms that are unknown or are difficult to express with rules or simple functions. Since energy functions are additive, terms learned by neural energy-based models can be naturally composed with those derived from physical knowledge.

In principle, neural networks have the ability to identify and represent non-additive higher order dependencies that might uncover features such as hydrogen bonding networks. Such features have been shown to have important roles in protein structure and function (Guo & Salahub, 1998; Redzic & Bowler, 2005; Livesay et al., 2008), and are important in protein design (Boyken et al., 2016). Incorporation of higher order terms has been an active research area for energy function design (Maguire et al., 2018).

Evaluations of molecular energy functions have used as a measure of fidelity, the ability to identify native side chain configurations (rotamers) from crystal structures where the ground-truth configuration has been masked out (Jacobson et al., 2002; Bower et al., 1997). Leaver-Fay et al. (2013) introduced a set of benchmarks for the Rosetta energy function that includes the task of rotamer recovery. In the benchmark, the ground-truth configuration of the side chain is masked and rotamers (possible configurations of the side chain) are sampled and evaluated within the surrounding molecular context (the rest of the atoms in the protein structure not belonging to the side chain). The energy function is scored by comparing the lowest-energy rotamer (as determined by the energy function) against the rotamer that was observed in the empirically-determined crystal structure.

This work takes an initial step toward fully learning an atomic-resolution energy function from data. Prediction of native rotamers from their context within a protein is a restricted problem setting for exploring how neural networks might be used to learn an atomic-resolution energy function for protein design. We compare the model to the Rosetta energy function, as detailed in Leaver-Fay et al. (2013), and find that on the rotamer recovery task, deep learning-based models obtain results approaching the performance of Rosetta. We investigate the outputs and representations of the model toward understanding its representation of molecular energies and exploring relationships to physical properties of proteins.

Our results open for future work the more general problem settings of combinatorial side chain optimization for a fixed backbone (Tuffery et al., 1991; Holm & Sander, 1992) and the inverse folding problem (Pabo, 1983) – the recovery of native sequences for a fixed backbone – which has also been used in benchmarking and development of molecular energy functions for protein design (Leaver-Fay et al., 2013).

## 2 BACKGROUND

**Protein conformation** Proteins are linear polymers composed of an alphabet of twenty canonical amino acids (residues), each of which shares a common backbone moiety responsible for formation of the linear polymeric backbone chain, and a differing side chain moiety with biochemical properties that vary from amino acid to amino acid. The energetic interplay of tight packing of side chains within the core of the protein and exposure of polar residues at the surface drives folding of proteins into stable molecular conformations (Richardson & Richardson, 1989; Dill, 1990).

The conformation of a protein can be described through two interchangeable coordinate systems. Each atom has a set of spatial coordinates, which up to an arbitrary rotation and translation of all coordinates describes a unique conformation. In the internal coordinate system, the conformation is described by a sequence of rigid-body motions from each atom to the next, structured as a kinematic tree. The major degrees of freedom in protein conformation are the dihedral rotations (Richardson & Richardson, 1989), about the backbone

bonds termed *phi* ($\phi$) and *psi* ($\psi$) angles, and the dihedral rotations about the side chain bonds termed *chi* ($\chi$) angles.

Within folded proteins, the side chains of amino acids preferentially adopt configurations that are determined by their molecular structure. A relatively small number of configurations separated by high energetic barriers are accessible to each side chain (Janin et al., 1978). These configurations are called *rotamers*. In Rosetta and other protein design methods, rotamers are commonly represented by libraries that estimate a probability distribution over side chain configurations, conditioned on the backbone $\phi$ and $\psi$ torsion angles. We use the Dunbrack library (Shapovalov & Dunbrack Jr, 2011) for rotamer configurations.

**Energy-based models** A variety of methods have been proposed for learning distributions of high-dimensional data, e.g. generative adversarial networks (Goodfellow et al., 2014) and variational autoencoders (Kingma & Welling, 2013). In this work, we adopt energy-based models (EBMs) (Dayan et al., 1995; Hinton & Salakhutdinov, 2006; LeCun et al., 2006). This is motivated by their simplicity and scalability, as well as their compelling results in other domains, such as image generation (Du & Mordatch, 2019).

In EBMs, a scalar parametric energy function $E_\theta(x)$ is fit to the data, with $\theta$ set through a learning procedure such that the energy is low in regions around $x$ and high elsewhere. The energy function maps to a probability density using the Boltzmann distribution: $p_\theta(x) = \exp(-E_\theta(x))/Z(\theta)$, where $Z = \int \exp(-E_\theta(x))\,dx$ denotes the partition function.

EBMs are typically trained using the maximum-likelihood method (ML), in which $\theta$ is adjusted to minimize $\mathrm{KL}(p_D(x)||p_\theta(x))$, the Kullback-Leibler divergence between the data and the model distribution. This corresponds to maximizing the log-likelihood of the data under the model:

$$L_{\mathrm{ML}}(\theta) = \mathbb{E}_{x \sim p_D}[\log p_\theta(x)] = \mathbb{E}_{x \sim p_D}[E_\theta(x) - \log Z(\theta)]$$

Following Carreira-Perpinan & Hinton (2005), the gradient of this objective can be written as:

$$\nabla_\theta L_{\mathrm{ML}} \approx \mathbb{E}_{x^+ \sim p_D}[\nabla_\theta E_\theta(x^+)] - \mathbb{E}_{x^- \sim p_\theta}[\nabla_\theta E_\theta(x^-)]$$

Intuitively, this gradient decreases the energy of samples from the data distribution $x^+$ and increases the energy of samples drawn from the model $x^-$. Sampling from $p_\theta$ can be done in a variety of ways, such as Markov chain Monte Carlo or Gibbs sampling (Hinton & Salakhutdinov, 2006), possibly accelerated using Langevin dynamics (Du & Mordatch, 2019). Our method uses a simpler scheme to approximate $\nabla_\theta L_{\mathrm{ML}}$, detailed in Section 3.4.

## 3 METHOD

Our goal is to score molecular configurations of the protein side chains given a fixed target backbone structure. We define an architecture for an energy-based model and describe its training procedure.

### 3.1 MODEL

The model calculates scalar functions, $f_\theta(A)$, of size-$k$ subsets, $A$, of atoms within a protein.

**Selection of atom subsets** In our experiments, we choose $A$ to be nearest-neighbor sets around the residues of the protein and set $k = 64$. For a given residue, we construct $A$ to be the $k$ atoms that are nearest to the residue's beta carbon.

**Atom input representations** Each atom in $A$ is described by its 3D Cartesian coordinates and categorical features: (i) the identity of the atom (N, C, O, S); (ii) an ordinal label of the atom in the side chain (i.e. which specific carbon, nitrogen, etc. atom it is in the side chain) and (iii) the amino acid type (which of the 20 types of amino acids the atom belongs to). The coordinates are normalized to have zero mean across the $k$

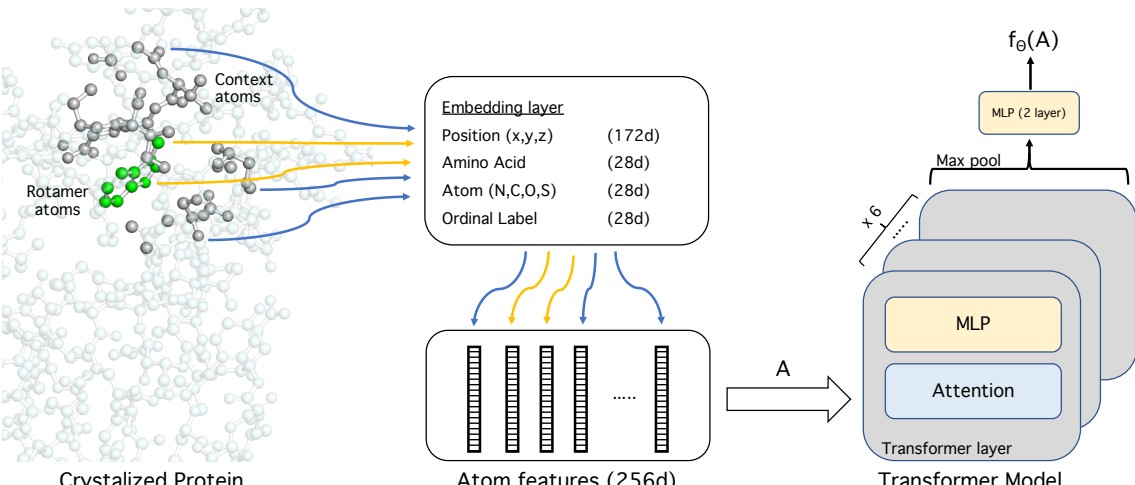

Figure 1: Overview of the model. The model takes as input a set of atoms, $A$, consisting of the rotamer to be predicted (shown in green) and surrounding atoms (shown in dark grey). The Cartesian coordinates and attributes of each atom are embedded. The set of embeddings is processed by Transformer blocks, and the final hidden representations are pooled over the atoms to produce a vector. The vector is passed through a two-layer MLP to output a scalar energy value, $f_\theta(A)$.

atoms. Each categorical feature is embedded into 28 dimensions, and the spatial coordinates are projected into 172 dimensions[2], which are then concatenated into a 256-dimensional atom representation. The parameters for the input embeddings and projections of spatial information are learned via training. During training, a random rotation is applied to the coordinates in order to encourage rotational invariance of the model. For visualizations, a fixed number of random rotations (100) is applied and the results are averaged.

**Architecture** In our proposed approach, $f_\theta(A)$ takes the form of a Transformer model (Vaswani et al., 2017) that processes a set of atom representations. The self-attention layers allow each atom to attend to the representations of other atoms in the set, modeling the energy of the molecular configuration as a non-linear interaction of single, pairwise, and higher-order interactions between the atoms. The final hidden representations of the Transformer are pooled across the atoms to produce a single vector, which is finally passed to a two-layer multilayer perceptron (MLP) that produces the scalar output of the model. Figure 1 illustrates the model.

For all experiments, we use a 6-layer Transformer with embedding dimension of 256 (split over 8 attention heads) and feed-forward dimension of 1024. The final MLP contains 256 hidden units. The models are trained without dropout. Layer normalization (Ba et al., 2016) is applied before the attention blocks.

## 3.2 PARAMETERIZATION OF PROTEIN CONFORMATIONS

The structure of a protein can be represented by two parameterizations: (1) absolute Cartesian coordinates of the set of atoms, and (2) internal coordinates of the atoms encoded as a set of in-plane/out-of-plane rotations and displacements relative to each atom's reference frame. Out-of-plane rotations are parameterized by $\chi$ angles which are the primary degrees of freedom in the rotamer configurations. The coordinate systems are interchangeable.

---

[2]The high dimensionality of the spatial projection was important to ensure a high weighting on the spatial coordinates, which proved necessary for the model to train reliably.

### 3.3 Usage as an energy function

We specify our energy function $E_\theta(x, c)$ to take an input set composed of two parts: (1) the atoms belonging to a rotamer to be predicted, $x$, and (2) the atoms of the surrounding molecular context, $c$. The energy function is defined as follows:

$$E_\theta(x, c) = f_\theta(A(x, c))$$

where $A(x, c)$ is the set of embeddings from $k$ atoms nearest to the rotamer's beta carbon.

### 3.4 Training and loss functions

In all experiments, the energy function is trained to learn the conditional distribution of the rotamer given its context by approximately maximizing the log likelihood of the data.

$$\mathcal{L}(\theta) = -E_\theta(x, c) - \log Z_\theta(c)$$

To estimate the partition function, we note that:

$$\log Z_\theta(c) = \log \int e^{-E_\theta(x, c)} dx = \log(\mathbb{E}_{q(x|c)}[\frac{e^{-E_\theta(x, c)}}{q(x|c)}])$$

for some importance sampler $q(x|c)$. Furthermore, if we assume $q(x|c)$ is uniformly distributed on supported configurations, we obtain a simplified maximum likelihood objective given by

$$\mathcal{L}(\theta) = -E_\theta(x, c) - \log(\mathbb{E}_{q(x^i|c)}[e^{-E_\theta(x^i, c)}])$$

for some context dependent importance sampler $q(x|c)$. We choose our sampler $q(x|c)$ to be an empirically collected rotamer library (Shapovalov & Dunbrack Jr, 2011) conditioned on the amino acid identity and the backbone $\phi$ and $\psi$ angles. We write the importance sampler as a function of atomic coordinates which are interchangeable with the angular coordinates in the rotamer library. The library consists of lists of means and standard deviations of possible $\chi$ angles for each 10 degree interval for both $\phi$ and $\psi$. We sample rotamers uniformly from this library, given by a continuous $\phi$ and $\psi$, by sampling from a weighted mixture of Gaussians of $\chi$ angles at each of the four surrounding bins, with weights given by distance to the bins via bilinear interpolation. Every candidate rotamer at each bin is assigned uniform probability. To ensure our context dependent importance sampler effectively samples high likelihood areas in the model, we further add the real context as a sample from $q(x|c)$.

**Training setup** Models were trained for 180 thousand parameter updates using 32 NVIDIA V100 GPUs, a batch size of 16,384, and the Adam optimizer ($\alpha = 2 \cdot 10^{-4}$, $\beta_1 = 0.99$, $\beta_2 = 0.999$). We evaluated training progress using a held-out 5% subset of the training data as a validation set.

## 4 Experiments

### 4.1 Datasets

We constructed a curated dataset of high-resolution PDB structures using the CullPDB database, with the following criteria: resolution finer than 1.8 Å; sequence identity less than 90%; and R value less than 0.25 as defined in Wang & R. L. Dunbrack (2003). To test the model on rotamer recovery, we use the test set of structures from Leaver-Fay et al. (2013). To prevent training on structures that are similar to those in the test set, we ran BLAST on sequences derived from the PDB structures and removed all train structures with more than 25% sequence identity to sequences in the test dataset. Ultimately, our train dataset consisted of 12,473 structures and our test dataset consisted of 129 structures.

| Model | Avg | Buried | Surface |
|-------|-----|--------|---------|
| Rosetta score12 (rotamer-trials) | 72.2 (72.6) | - | - |
| Rosetta ref2015 (rotamer-trials) | 73.6 | - | - |
| Atom Transformer | 70.4 | 87.0 | 58.3 |
| Atom Transformer (ensemble) | 71.5 | 89.2 | 59.9 |

Table 1: Rotamer recovery of energy functions under the discrete rotamer sampling method detailed in Section 4.2.1. Parentheses denote value reported by Leaver-Fay et al. (2013).

## 4.2 BASELINES

We compare to three baseline neural network architectures: a fully-connected network, the architecture for embedding sets in the set2set paper (Vinyals et al., 2015); and a graph neural network (Veličković et al., 2017). All models have around 10 million parameters. Details of the baseline architectures are given in Appendix A.1.2.

Results are also compared to Rosetta. We ran Rosetta using score12 and and ref15 energy functions using the rotamer trials and rtmin protocols with default settings.

### 4.2.1 EVALUATION

For the comparison of the model to Rosetta in Table 1, we reimplement the sampling scheme that Rosetta uses for rotamer trials evaluation. We take discrete samples from the rotamer library, with bilinear interpolation of the mean and standard deviations using the four grid points surrounding the backbone $\phi$ and $\psi$ angles for the residue. We take discrete samples of the rotamers at $\mu$, except that for buried residues we sample $\chi_1$ and $\chi_2$ at $\mu$ and $\mu \pm \sigma$ as was done in Leaver-Fay et al. (2013). We define buried residues to have $\geq 24$ $C_\beta$ neighbors within 10Å of the residue's $C_\beta$ ($C_\alpha$ for glycine residues). For buried positions we accumulate rotamers up to 98% of the distribution, and for other positions the accumulation is to 95%. We score a rotamer as recovered correctly if all $\chi$ angles are within $20°$ of the ground-truth residue.

We also use a continuous sampling scheme which approximates the empirical conditional distribution of the rotamers using a mixture of Gaussians with means and standard deviations computed by bilinear interpolation as above. Instead of sampling discretely, the component rotamers are sampled with the probabilities given by the library, and a sample is generated with the corresponding mean and standard deviation. This is the same sampling scheme used to train models, but with component rotamers now weighted by probability as opposed to uniform sampling.

## 4.3 ROTAMER RECOVERY RESULTS

Table 1 directly compares our EBM model (which we refer to as the Atom Transformer) with two versions of the Rosetta energy function. We run Rosetta on the set of 152 proteins from the benchmark of Leaver-Fay et al. (2013). We also include published performance on the same test set from Leaver-Fay et al. (2013). As discussed above, comparable sampling strategies are used to evaluate the models, enabling a fair comparison of the energy functions. We find that a single model evaluated on the benchmark performs slightly worse than both versions of the Rosetta energy function. An ensemble of 10 models improves the results.

Table 2 evaluates the performance of the energy function under alternative sampling strategies with the goal of optimizing recovery rates. We indicate performance of the Rosetta energy function on recovery rates using the rtmin protocol for continuous minimization. We evaluate the learned energy function with the continuous sampling from a mixture of Gaussians conditioned on the $\phi/\psi$ settings of the backbone angles as detailed

| Model | Avg | Buried | Surface |
|---|---|---|---|
| Fully-connected | 39.1 | 54.4 | 30.0 |
| Set2set | 43.2 | 60.3 | 31.7 |
| GraphNet | 69.0 | 94.3 | 54.2 |
| Atom Transformer | 73.1 | 91.1 | 58.3 |
| Atom Transformer (ensemble) | 74.1 | 91.2 | 59.5 |
| Rosetta score12 (rt-min) | 75.4 (74.2) | - | - |
| Rosetta ref2015 (rt-min) | 76.4 | - | - |

Table 2: Rotamer recovery of energy functions under continuous optimization schemes. Rosetta continuous optimization is performed with the rtmin protocol. Parentheses denote value reported by Leaver-Fay et al. (2013).

| Amino Acid | R | K | M | I | L | S | T | V |
|---|---|---|---|---|---|---|---|---|
| Atom Transformer | 37.2 | 31.7 | 53.0 | 93.3 | 82.6 | 79.0 | 96.5 | 94.0 |
| Rosetta score12 | 26.7 | 31.7 | 49.6 | 85.4 | 87.5 | 72.5 | 92.6 | 94.3 |
| Amino Acid | N | D | Q | E | H | W | F | Y |
| Atom Transformer | 67.4 | 76.0 | 40.8 | 49.8 | 65.5 | 83.5 | 80.3 | 77.6 |
| Rosetta score12 | 56.8 | 60.4 | 30.7 | 33.6 | 55.0 | 85.0 | 85.4 | 82.9 |

Table 3: Comparison of rotamer recovery rates by amino acid between Rosetta and the ensembled energy-based model under discrete rotamer sampling. The model appears to perform well on polar amino acids glutamine, serine, asparagine, and threonine, while Rosetta performs better on larger amino acids phenylalanine, tyrosine, and tryptophan and the common amino acid, leucine. The numbers reported for Rosetta are from Leaver-Fay et al. (2013).

above. We find that with ensembling the model performance is close to that of the Rosetta energy functions. We also compare to three baselines for embedding sets with similar numbers of parameters to the Atom Transformer model and find that they have weaker performance.

Buried residues are more constrained in their configurations by tight packing of the side chains within the core of the protein. In comparison, surface residues are more free to vary. Therefore we also report performance separately on both categories. We find that the ensembled Atom Transformer has a 91.2% rotamer recovery rate for buried residues, compared to 59.5% for surface residues.

Table 3 reports recovery rates by residue comparing the Rosetta score12 results reported in Leaver-Fay et al. (2013) to the Atom Transformer model using the Rosetta discrete sampling method. The Atom Transformer model appears to perform well on smaller rotameric amino acids as well as polar amino acids such as glutamate/aspartate while Rosetta performs better on larger amino acids like phenylalanine and tryptophan and more common ones like leucine.

## 4.4 VISUALIZING ENERGIES

In this section, we visualize and understand how the Atom Transformer models the energy of rotamers in their native contexts. We explore the response of the model to perturbations in the configuration of side chains away from their native state. We retrieve all protein structures in the test set and individually perturb rotameric $\chi$ angles across the unit circle, plotting results in Figures 2, 3, and 4.

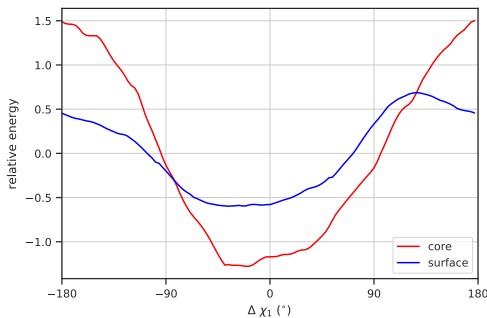

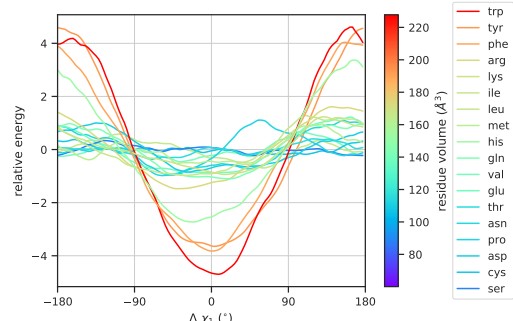

Figure 2: The energy function models distinct behavior between core and surface residues. Core residues are more sensitive to perturbations away from the native state in the $\chi_1$ torsion angle. On average, residues closer to the core have a steeper energy well.

Figure 3: There is a relation between the residue size and the depth of the energy well, with larger amino acids (e.g. Trp, Phe, Thr, Lys) having steeper wells.

**Core/Surface Energies**    Figure 2 shows that steeper response to variations away from the native state is observed for residues in the core of the protein (having $\geq 24$ contacting side chains) than for residues on the surface ($\leq 16$), consistent with the observation that buried side chains are tightly packed (Richardson & Richardson, 1989).

**Rotameric Energies**    Figure 3 shows a relation between the residue size and the depth of the energy well, with larger amino acids having steeper wells (more sensitive to perturbations). Furthermore Figure 4 shows that the model learns the symmetries of amino acids. We find that responses to perturbations of the $\chi_2$ angle for the residues Tyr, Asp, and Phe are symmetric about $\chi_2$. A 180° periodicity is observed, in contrast to the non-symmetric residues.

**Embeddings of Atom Sets**    Building on the observation of a relation between the depth of the residue and its response to perturbation from the native state, we ask whether core and surface residues are clustered within the representations of the model. To visualize the final hidden representation of the molecular contexts within a protein, we compute the final vector embedding for the 64 atom context around the carbon-$\beta$ atom (or for glycine, the carbon-$\alpha$ atom) for each residue. We find that a projection of these representations by t-SNE (Maaten & Hinton, 2008) into 2 dimensions shows a clear clustering between representations of core residues and surface residues. A representative example is shown in Figure 5.

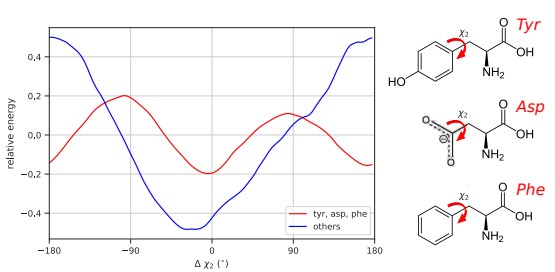

Figure 4: Note the periodicity for the amino acids Tyr, Asp, and Phe with terminal symmetry about $\chi_2$.

**Saliency Map**    The 10-residue protease-binding loop in a chymotrypsin inhibitor from barley seeds is highly structured due to the presence of backbone-backbone and backbone-sidechain hydrogen bonds in the same residue (Das, 2011). To visualize the dependence of the energy function on individual atoms, we compute the energy of the 64 atom context centered around the backbone carbonyl oxygen of residue 39 (isoleucine) in PDB: 2CI2 (McPhalen & James, 1987) and derive the gradients with respect to the input atoms. Figure 6 overlays the gradient magnitudes on the structure, indicating the model attends to both sidechain and backbone atoms, which participate in hydrogen bonds.

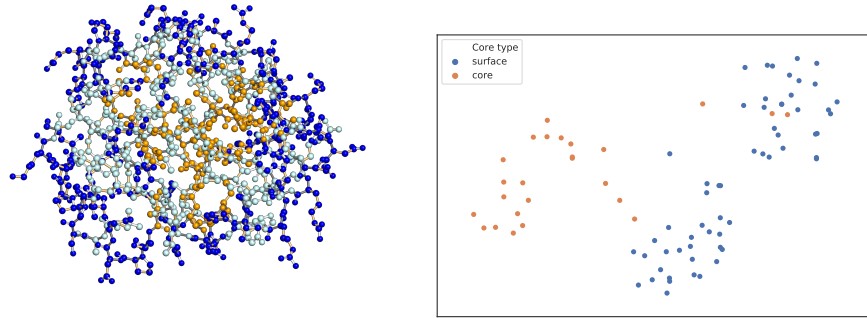

Figure 5: Left: 3-dimensional representation of CcmG reducing oxidoreductase (PDB ID 1KNG; Edeling et al., 2002), a protein from the test set. Atoms are colored dark blue (buried), orange (exposed), or neither (not colored). Right: t-SNE (Maaten & Hinton, 2008) projection of EBM hidden representation when focused on the alpha carbon atom for each residue in the hidden representation. In the embedding space, buried and surface residues are distinguished.

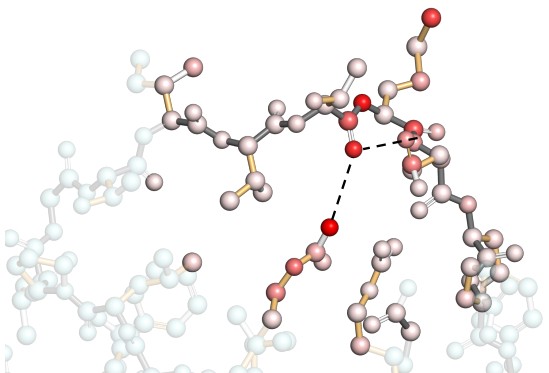

Figure 6: The model's saliency map applied to the test protein, serine proteinase inhibitor (PDB ID: 2CI2; McPhalen & James (1987)). The 64 atom context is centered on the carbonyl oxygen of residue 39 (isoleucine). Atoms in the context are labeled red with color saturation proportional to gradient magnitude (interaction strength). Hydrogen bonds with the carbonyl oxygen are shown by dotted lines.

## 5 RELATED WORK

Energy functions have been widely used in the modeling of protein conformations and the design of protein sequences and structures (Boas & Harbury, 2007). Rosetta, for example, uses a combination of physically motivated terms and knowledge-based potentials (Alford et al., 2017) to model proteins and other macromolecules.

Leaver-Fay et al. (2013) proposed optimizing the feature weights and parameters of the terms of an energy function for protein design; however their method used physical features designed with expert knowledge and data analysis. Our work draws on their development of rigorous benchmarks for energy functions, but in contrast automatically learns complex features from data.

Neural networks have also been explored for protein folding. Xu (2018) developed a deep residual network that predicts the pairwise distances between residues in the protein structure from evolutionary covariation information. Senior et al. (2018) used evolutionary covariation to predict pairwise distance distributions,

using maximization of the probability of the backbone structure with respect to the predicted distance distribution to fold the protein. Ingraham et al. (2018) proposed learning an energy function for protein folding by backpropagating through a differentiable simulator. AlQuraishi (2019) investigated predicting protein structure from sequence without using co-evolution.

Deep learning has shown practical utility in the related field of small molecule chemistry. Gilmer et al. (2017) achieved state-of-the-art performance on a suite of molecular property benchmarks. Similarly, Feinberg et al. (2018) achieved state-of-the-art performance on predicting the binding affinity between proteins and small molecules using graph convolutional networks. Mansimov et al. (2019) used a graph neural network to learn an energy function for small molecules. In contrast to our work, these methods operate over small molecular graphs and were not applied to large macromolecules, like proteins.

In parallel, recent work proposes that generative models pre-trained on protein sequences can transfer knowledge to downstream supervised tasks (Bepler & Berger, 2019; Alley et al., 2019; Yang et al., 2019; Rives et al., 2019). These methods have also been explored for protein design (Wang et al., 2018).

Generative models of protein structures have also been proposed for generating protein backbones (Anand & Huang, 2018) and for the inverse protein folding problem (Ingraham et al., 2019).

## 6 DISCUSSION

In this work we explore the possibility of learning an energy function of protein conformations at atomic resolution. We develop and evaluate the method in the benchmark problem setting of recovering protein side chain conformations from their native context, finding that a learned energy function nears the performance in this restricted domain to energy functions that have been developed through years of research into approximation of the physical forces guiding protein conformation and engineering of statistical terms.

The method developed here models sets of atoms and can discover and represent the energetic contribution of high order dependencies within its inputs. We find that learning an energy function from the data of protein crystal structures automatically discovers features relevant to computing molecular energies; and we observe that the model responds to its inputs in ways that are consistent with an intuitive understanding of protein conformation and energy.

Generative biology proposes that the design principles used by nature can be automatically learned from data and can be harnessed to generate and design new biological molecules and systems (Rives et al., 2019). High-fidelity generative modeling for proteins, operating at the level of structures and sequences, can enable generative protein design. To create new proteins outside the space of those discovered by nature, it is necessary to use design principles that generalize to all proteins. Huang et al. (2016) have argued that since the physical principles that govern protein conformation apply to all proteins, encoding knowledge of these physical and biochemical principles into an energy function will make it possible to design *de novo* new protein structures and functions that have not appeared before in nature.

Learning features from data with generative methods is a possible direction for realizing this goal to enable design in the space of sequences not visited by evolution. The generalization of neural energy functions to harder problem settings used in the protein design community, e.g. combinatorial side chain optimization (Tuffery et al., 1991; Holm & Sander, 1992), and inverse-folding (Pabo, 1983), is a direction for future work. The methods explored here have the potential for extension into these settings.

ACKNOWLEDGMENTS

We thank Kyunghyun Cho, Siddharth Goyal, Andrew Leaver-Fay, and Yann LeCun for helpful discussions. Alexander Rives was supported by NSF Grant #1339362.

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

## A.1 Appendix

### A.1.1 Pseudocode for the Training Algorithm

Pseudocode for the training algorithm is given below in Algorithm 1.

---

**Algorithm 1** Training Procedure for the EBM

---

**Input:** Rotamer library $q(x|c)$, Training set of proteins $D$
**for** Protein $d_i$ of $D$ **do**
   ▷ *Sample random amino acid from $d_i$*
   $R \sim d_i$
   ▷ *Set positive sample to 64 nearest neighbor atoms of carbon beta of R*
   $c^+ \leftarrow \text{NN}_{64}(R)$
   ▷ *Generate N negative samples from the rotamer library*
   $c^- \leftarrow q(x|c^+)$
   ▷ *Compute loss of model (logsumexp across all negative samples)*
   $L_{ml} = E(c^+; \theta) + \text{logsumexp}(-E(c^+; \theta), -E(c_0^-; \theta), -E(c_1^-; \theta), \ldots, -E(c_N^-; \theta))$
   ▷ *Minimization step of $L_{ml}$ using Adam optimizer*
   $\theta \leftarrow \theta - \nabla_\theta L_{ml}$
**end for**

---

### A.1.2 Model Architecture

Architectural descriptions are provided below for each the three neural network baselines as well as for the Atom Transformer. For Set2set models, we use 6 processing steps of computation. For graph networks, we add residual connections between each layer.

| Embed Each Atom to 256 Dim |
|:---:|
| Flatten |
| Dense $\rightarrow$ 1024 |
| $1024 \rightarrow 1024$ |
| $1024 \rightarrow 1024$ |
| ResBlock down 256 |
| Global Mean Pooling |
| Dense $\rightarrow$ 1 |

(a) Fully Connected Model

| Embed Each Atom to 256 Dim |
|:---:|
| Dense $\rightarrow$ 1024 |
| Repeat (6x): |
| LSTM 2048 Attention $2048 \rightarrow 128 \rightarrow 1$ |
| End Repeat |
| Dense $\rightarrow$ 1024 |
| $1024 \rightarrow 1$ |

(b) Set2Set Model (6 Permutation Invariant Blocks)

Figure A1: Architectures for Fully Connected and Set2Set Baselines

| |
| --- |
| Embed Each Atom to 512 Dim |
| Graph Attention Layer |
| Graph Attention Layer |
| Graph Attention Layer |
| Graph Attention Layer |
| Graph Attention Layer |
| Graph Attention Layer |
| Graph Attention Layer |
| Graph Attention Layer |
| Graph Attention Layer |
| Global Average Pooling |
| dense $\rightarrow$ 1 |

Figure A2: Graph Network Model Architecture (9 Graph Attention Blocks)

| |
| --- |
| Embed Each Atom to 256 Dim |
| Transformer Encoder Block (8 heads, feedforward dim 1024, 256 encoder dim) |
| Transformer Encoder Block (8 heads, feedforward dim 1024, 256 encoder dim) |
| Transformer Encoder Block (8 heads, feedforward dim 1024, 256 encoder dim) |
| Transformer Encoder Block (8 heads, feedforward dim 1024, 256 encoder dim) |
| Transformer Encoder Block (8 heads, feedforward dim 1024, 256 encoder dim) |
| Transformer Encoder Block (8 heads, feedforward dim 1024, 256 encoder dim) |
| Global Max Pooling |
| dense $\rightarrow$ 1 |

Figure A3: Atom Transformer Model (6 Transformer Encoder Blocks)

