# OpenReview forum: "Energy-based models for atomic-resolution protein conformations"
_ICLR.cc/2020/Conference — Accept (Spotlight)_

### Official Review · AnonReviewer2 · 2019-10-23
**Official Blind Review #2**

**Rating:** 8

**Review:**

The paper proposes an Energy-Based-Model (EBM) for scoring the possible configurations of amino acid side chain conformations in protein structures with known amino acid backbone structure. The energy of the side-chain conformation (the chi-angle) for a given amino acid in the structure is calculated as a function of a local neighbourhood of atoms (A), where each atom is embedded into a 256d vector using its cartesian coordinates, atom identity, atom side-chain position and amino acid identity. The model is trained using approximate likelihood where the model samples are generated using precalculated table (from literature) of possible Chi angles conformations conditioned on the back-bone amino acid identity and back-bone angles. The results seem comprehensive comparing the transformer based energy function parameterization with two sensible baselines as well as the Rosetta energy function  which is the de facto standard tool for these types of calculations. Using rotamer recovery accuracy as the benchmark measure the empirical results are close to performance as the Rosetta energy model however always slightly worse. Further visualizations of the energy levels for different Chi angles seems to support that the learned energy function captures well known characteristics of the rotamer configuration energy landscape.

Score:
Overall I think that the paper is solid and tackles an interesting problem of learning energy functions for physical systems from data. The experimental results are comprehensive and mostly supports the claims made in the paper. My main (slightly minor) concern about the paper is the obsession with discarding years of learned knowledge about handcrafted energy functions with fully learned functions. It seems to me that combining domain knowledge with a learned model should close the last small gap (and presumably surpass) the performance of e.g Rosetta. Combining I think the paper should be accepted at the ICLR.

Comment/Questions:

Motivation
Q1.1) Paper motivation: The paper several times seems to suggest that energy functions derived from physical knowledge are problematic (e.g. abstract). For many physical systems I don’t think this is true and would like the author's comments on why learned energy functions are preferable - arguably they can only capture properties present in the data and not any prior knowledge?

Q1.2) Given that the paper is motivated by fully learning an energy function from data without injecting any prior knowledge i would like the authors comment on the construction of the q(x|c) distribution. This is based on essentially a contingency table between Phi/Psi/Chi angles and to me seems like injecting prior knowledge about physically possible rotamer configurations into the learned procedure?

Method / Experimental Results:
Q2.1) With respect to the primary results in table 1) and table 2). The authors claim comparable results to the Rosetta Energy function (page 5. Sec 4.3, page 9, sec 6). However the experimental results are always slightly worse than the Rosetta energy function and I (strongly) suggest that the authors rephrase those statements to reflect that.

Q2.2 With Respect to Table 3) Firstly, Why are the only results for 16 amino acids in the table ? Secondly, just naively counting the Atom Transformer are better than Rosetta in 11 of 16 amino acids - this seems slightly at odds with the main result where the Atom Transformer is performing slightly worse than Rosetta?

Clarity
Q3): The notation in section 3 and 4 is slightly confusing. Especially I think q(x|c) is slightly misleading since it, (to my understanding) is a conditional distribution over Chi, conditioned on Psi/Phi/AminoAcid and not rotamer (x) and surrounding molecular context (c).

**Experience Assessment:**

I have published one or two papers in this area.

**Review Assessment: Checking Correctness Of Derivations And Theory:**

I assessed the sensibility of the derivations and theory.

**Review Assessment: Checking Correctness Of Experiments:**

I carefully checked the experiments.

**Review Assessment: Thoroughness In Paper Reading:**

I read the paper thoroughly.

---

> ### Author Response · Authors · 2019-11-15
> **Response**
>
> We thank the reviewer for constructive criticism and questions. This feedback has been very helpful and we've made a number of alterations to the text in response.
>
> >The paper several times seems to suggest that energy functions derived from physical knowledge are problematic (e.g. abstract)
>
> We respectfully disagree with the reviewer on this point. The abstract states: “The model is trained solely on crystallized protein data. By contrast, existing approaches for scoring conformations use energy functions that incorporate knowledge of physical principles and features that are the complex product of several decades of research and tuning.” To the best of our knowledge, this is an accurate statement about current energy functions. Our argument is not about the absolute performance of physical or learned approaches, but rather the ability of learning-based methods to approach the performance of knowledge-based methods with less development effort, thus indicating potential for this direction.
>
> >My main (slightly minor) concern about the paper is the obsession with discarding years of learned knowledge about handcrafted energy functions with fully learned functions. It seems to me that combining domain knowledge with a learned model should close the last small gap (and presumably surpass) the performance of e.g Rosetta.
>
> We strongly agree that the possibility of combining energy functions and features learned by neural networks with existing energy functions that incorporate domain knowledge is a promising direction for future work. In the introduction we write “Furthermore, since energy functions are additive, terms learned by neural energy-based models can be naturally composed with those proposed by expert knowledge.” We note that this possibility parallels (and might be seen as an extension of) the long line of work combining statistical potential terms with physical energy terms.
>
> We revise the above statement as follows: “Since energy functions are additive, terms learned by neural energy-based models can be naturally composed with those derived from physical knowledge.”
>
> >Given that the paper is motivated by fully learning an energy function from data without injecting any prior knowledge i would like the authors comment on the construction of the q(x|c) distribution. This is based on essentially a contingency table between Phi/Psi/Chi angles and to me seems like injecting prior knowledge about physically possible rotamer configurations into the learned procedure?
>
> The rotamer library we use [1] has been fit from data. We note that this library is only used to sample rotamer configurations and that the energy function does not directly include terms or features analytically derived from rotamer probabilities.
>
> We also noticed in the related work section where we discuss the work of Leaver-Fay et al. [2] we had stated “our method automatically learns complex features from data without prior knowledge.” We apologize if this contributes to an impression that a major concern of the paper is learning without prior knowledge. We revise this to “our method automatically learns complex features from data.”
>
> >However the experimental results are always slightly worse than the Rosetta energy function and I (strongly) suggest that the authors rephrase those statements to reflect that.
>
> We thank the reviewer for pointing out this concern which we agree is valid, and we have revised the paper accordingly to state “our model achieves performance close to that of the Rosetta energy function” throughout.
>
> >Why are the only results for 16 amino acids in the table ?
>
> Although we state in the main text that the numbers reported in Table 3 for Rosetta are from the paper of Leaver-Fay et al. [2], we overlooked mentioning this in the caption to the table. We've rectified this. We only present results for those 16 amino acids because they are the only ones included in the Leaver-Fay et al. paper.
>
> >Secondly, just naively counting the Atom Transformer are better than Rosetta in 11 of 16 amino acids - this seems slightly at odds with the main result where the Atom Transformer is performing slightly worse than Rosetta?
>
> We also agree it is surprising that our method performs better on many of the amino acids. We observe that our performance is worse on some of the more common amino acids.
>
> >The notation in section 3 and 4 is slightly confusing
>
> The angular coordinates and Cartesian coordinates are interchangeable. Since the model takes as input Cartesian coordinates of the atoms, we find it more expressive and convenient to write the importance distribution in this form. We have revised the paper to clarify this.
>
> ---
> [1] Shapovalov and Dunbrack. "A smoothed backbone-dependent rotamer library for proteins derived from adaptive kernel density estimates and regressions." (2011)
>
> [2] Andrew Leaver-Fay, et al. "Scientific benchmarks for guiding macromolecular energy function improvement." (2013)

---

> > ### Comment · AnonReviewer2 · 2019-11-15
> > **Response**
> >
> > Thanks for addressing my comments - as reflected in my score ( I'll stay by my original accept-score)  I find this paper interesting and hope to see it at ICLR2020.

---

### Official Review · AnonReviewer1 · 2019-10-24
**Official Blind Review #1**

**Rating:** 8

**Review:**

Summary of the paper

The authors propose a predictive model based on the energy based model that uses a Transformer architecture for the energy function. It accepts as input an atom and its neighboring atoms and computes an energy for their configuration. The input features include representations of physical properties (atom identity, atom location within a side chain and amino-acid type) and spatial coordinates (x, y, z). A set of 64 atoms closest to the beta carbon of the target residue are selected and each is projected to a 256-dimensional vector. The predictive model computes an energy for the configuration of these 64 atoms surrounding a residue under investigation. The model is reported to achieve a slightly worse but comparable performance to the Rosetta energy function, the state-of-the-art method widely used in protein structure prediction and design. The authors investigate model’s outputs and hidden representations and conclude that it captures physicochemical properties relevant to the protein energy in general.

Strengths
+ A very interesting contribution to structural biology with a non-trivial application of deep learning.
+ The study is accompanied with structural biology interpretation of the designed energy predictor. Usually similar studies do not attempt and are limited with prediction rates and mathematical analysis.
+ Appropriate background is provided for non-experts in protein structural biology. The paper is clear and well written.
+ An adequate literature review is provided relative to the problem. Bird-eye view is given for future direction with justified optimism.

Weaknesses:
- The study claims to provide an energy predictor in general while training is performed on rotamer recovery of a single amino acid. These claims should be downplayed and what was shown at most is that this predictor can be applied to predict energy for a restricted problem of the rotamer recovery task.
- A reader gets a wrong impression at the beginning of the manuscript that the study tries to solve general and classical rotamer prediction for an entire protein. It becomes clear only at the very end that the study does not try to resolve all rotamer conformations in a protein, it can only predict one rotamer at the time given the atoms surrounding the target residue are correct. This should be explained in the beginning that the study does not attempt combinatorial side chain optimization for a fixed backbone;
- In-depth description of the neural network is required at least in supplemental materials, so that the study could be replicated. Ideally, a working application and training protocol code would be available.
- No techniques against overtraining are discussed. How was the model validated?

Small issues:
* function f_theta(A) is used before it is introduced.
* noun is missing: using our trained with deep learning.
* articles missing: that vary from amino acid to amino acid.
* KL abbreviation not explained
* MCMC abbreviation not explained
* wrong artile:  that processes the set of atom representations.
* misprint: resolution fine r than 1.8
* wrong word, has to be “lower”: sequence identity greater ˚
* everyday word: break out
* abbreviation not explained: t-SNE
* misprint case: Similarly, In contrast to our work


**Experience Assessment:**

I have read many papers in this area.

**Review Assessment: Checking Correctness Of Derivations And Theory:**

I assessed the sensibility of the derivations and theory.

**Review Assessment: Checking Correctness Of Experiments:**

I assessed the sensibility of the experiments.

**Review Assessment: Thoroughness In Paper Reading:**

I read the paper at least twice and used my best judgement in assessing the paper.

---

> ### Author Response · Authors · 2019-11-15
> **Response**
>
> We thank the reviewer for thoughtful and detailed comments.
>
> >The study claims to provide an energy predictor in general while training is performed on rotamer recovery of a single amino acid. These claims should be downplayed and what was shown at most is that this predictor can be applied to predict energy for a restricted problem of the rotamer recovery task.
>
> >A reader gets a wrong impression at the beginning of the manuscript that the study tries to solve general and classical rotamer prediction for an entire protein. It becomes clear only at the very end that the study does not try to resolve all rotamer conformations in a protein, it can only predict one rotamer at the time given the atoms surrounding the target residue are correct. This should be explained in the beginning that the study does not attempt combinatorial side chain optimization for a fixed backbone;
>
> We apologize that this was not clear -- we endeavored to make explicit that this paper only addresses the rotamer recovery problem and not the more general combinatorial side chain optimization and sequence recovery problems. For example, the abstract states: “To evaluate our model, we benchmark on the rotamer recovery task, a restricted problem setting used to evaluate energy functions for protein design.” And in the introduction: “Our results serve as a gateway for progress on the more general problem settings of combinatorial side chain optimization for a fixed backbone (Tuffery et al., 1991; Holm & Sander, 1992) and the inverse folding problem (Pabo, 1983)…”.
>
> However, we acknowledge that the wording could be clearer and have rewritten them as follows. We hope that this helps.
>
> “To evaluate our model, we benchmark on the rotamer recovery task, the problem of predicting the conformation of a side chain from its context within a protein structure, which has been used to evaluate energy functions for protein design.”
>
> “Our results open for future work the more general problem settings of combinatorial side chain optimization for a fixed backbone (Tuffery et al., 1991; Holm & Sander, 1992) and the inverse folding problem (Pabo, 1983)…”
>
> >In-depth description of the neural network is required at least in supplemental materials, so that the study could be replicated. Ideally, a working application and training protocol code would be available.
>
> We intend to make the code available before the ICLR conference is held. We’ve also added an appendix with a description of the network, baselines, and training algorithm.
>
> >No techniques against overtraining are discussed. How was the model validated?
>
> We added a brief description in the text. In short, we used a held-out 5% subset of the training data as a validation set.
>
> >Small issues
>
> We appreciate very much the careful attention to proofing errors in the text and we have made the corrections identified.

---

### Official Review · AnonReviewer3 · 2019-10-28
**Official Blind Review #2477**

**Rating:** 6

**Review:**

The paper proposes an energy based model (learned using the transformer architecture) to learn protein conformation from data describing protein crystal structure (rather than one based on knowledge of physical and biochemical principles). To test the efficacy of their method, the authors perform the task of identifying rotamers (native side-chain configurations) from crystal structures, and compare their method to the Rosetta energy function which is the current state-of-the-art.

+ Presents an important and emerging application of neural networks
+ Relatively clearly written, e.g. giving good background on protein conformation and related things to a machine learning audience

-The baselines that are compared to are set2set and Rosetta. Are there newer baselines the authors can compare to? For example, the related works section discusses some recent works.
-I am having a hard time interpreting the numbers in the experimental results (Table 1 and Table 2). I understand that they are rotamer recovery rates and that the max would be 100%. But I can’t tell what amount of difference is significant.
-Figure 1 should have a caption clarifying notation etc.


**Experience Assessment:**

I do not know much about this area.

**Review Assessment: Checking Correctness Of Derivations And Theory:**

I assessed the sensibility of the derivations and theory.

**Review Assessment: Checking Correctness Of Experiments:**

I carefully checked the experiments.

**Review Assessment: Thoroughness In Paper Reading:**

I read the paper at least twice and used my best judgement in assessing the paper.

---

> ### Author Response · Authors · 2019-11-15
> **Response**
>
> We thank the reviewer for the comments and helpful questions.
>
> >Are there newer baselines the authors can compare to?
>
> Rosetta is considered to be state of the art for protein energy functions. Additionally we’ve updated the paper to include an additional baseline in the form of a graph neural network. Its performance is below that of the Transformer but above that of the other baselines.
>
> >For example, the related works section discusses some recent works
>
> There are a number of possible architectures that might make sense for this problem, but to our knowledge none of them have been studied previously for rotamer recovery. We think it is a very interesting direction for future work to explore what inductive biases and alternative model architectures might improve performance and generalization.
>
> >I am having a hard time interpreting the numbers in the experimental results (Table 1 and Table 2). I understand that they are rotamer recovery rates and that the max would be 100%.
>
> We’re not aware of an estimate of the theoretically attainable maximum performance for this task, but we expect it to be well below 100%. This is because the side chains are free to adopt a variety of conformations when they are less constrained - especially at the surface of the protein where there is not tight packing. We expect that the Rosetta numbers represent very good performance (since Rosetta has been used successfully to design new functional proteins).
>
> >I can’t tell what amount of difference is significant
>
> We don’t have a principled way to determine error bars for the Rosetta results. We note that the gap in performance between score12 and ref2015 (the newer version of the energy function) is approximately 1-1.4% for rotamer trials and 1-2.2% for rt-min. These performance gains are regarded as being significant, thus providing some objective scale for our results. We have revised the paper to state our model achieves performance close to that of the Rosetta energy function rather than comparable to Rosetta.
>
> >Figure 1 should have a caption clarifying notation etc.
>
> We agree this is an oversight in the original draft and we’ve added a caption to Figure 1.

---

### Author Response · Authors · 2019-11-15
**Summary response**

We would like to express sincere appreciation to the reviewers for insightful and detailed comments and for the constructive nature of their feedback. We appreciate their interest in our work and the unanimously positive evaluation.

---

### Decision · Program_Chairs · 2019-12-19

**Decision:**

Accept (Spotlight)

**Comment:**

The paper proposes a data-driven approach to learning atomic-resolution energy functions. Experiment results show that the proposed energy function is similar to the state-of-art method (Rosetta) based on physical principles and engineered features.

The paper addresses an interesting and challenging problem. The results are very promising. It is a good showcase of how ML can be applied to solve an important application problem.

For the final version, we suggest that the authors can tune down some claims in the paper to fairly reflect the contribution of the work.